# Exploring Ocean Floor Geodiversity in Relation to Mineral Resources in the Southwest Pacific Ocean

**Arie Christoffel Seijmonsbergen** [1,*] , **Sanne Valentijn** [2] , **Lisan Westerhof** [3] **and Kenneth Frank Rijsdijk** [1]

1   Institute for Biodiversity and Ecosystem Dynamics, University of Amsterdam,
    1090 GE Amsterdam, The Netherlands; k.f.rijsdijk@uva.nl
2   Water Systems and Global Change Group, Wageningen University & Research,
    6700 AA Wageningen, The Netherlands; sanne.valentijn@kpnmail.nl
3   Institute for Interdisciplinary Studies, University of Amsterdam, 1090 GE Amsterdam, The Netherlands;
    lisanwesterhof@hotmail.com
*   Correspondence: a.c.seijmonsbergen@uva.nl; Tel.: +31-20-525-8137

**Abstract:** The future extraction of mineral resources may irreversibly damage ocean floor geodiversity in the Southwest Pacific Ocean. Therefore, understanding of the spatial distribution of ocean floor geodiversity in relation to mineral resources is important. For that purpose, we first developed a geodiversity index map of the western Pacific Ocean including spatial information of openly available digital layers of four components: seafloor geomorphology, sediment thickness, bathymetry and seafloor roughness. Second, we analysed how these components contributed to the geodiversity index. Finally, correlations between three mineral resources (seafloor massive sulphides, polymetallic nodules and cobalt-rich crusts) and the geodiversity index, its components, and the ocean floor age were calculated. The results showed that the ocean floor environment and the time necessary for the formation of the three mineral resources were predominantly related to the bathymetry component and the age of the ocean floor, and to a lesser extent to the seafloor roughness, geomorphology and sediment thickness components. We conclude that the ocean floor geodiversity index and its components contribute to an improved understanding of the spatial distribution of abiotic seafloor diversity and can be optimized by using higher resolution data. We suggest that ocean floor geodiversity could be considered in future resource extraction to support responsible mining and help limit environmental damage.

**Keywords:** seabed mining; Pacific Ocean; ocean floor geodiversity; bathymetry; cobalt-rich crusts; polymetallic nodules; seafloor massive sulphides





## 1. Introduction

The geodiversity of the ocean floor, which makes up more than seventy percent of the Earth's geodiversity, remains largely unexplored. It is widely recognized that geodiversity provides key geosystem services [1,2], has a well-defined role in establishing UNESCO Global Geoparks [3], contains crucial elements for preserving cultural heritage [1], has great educational potential [4], and has proven relations to biodiversity and habitats in a variety of environments [5,6]. Geodiversity contributes to natural capital and provides benefits to many (provisional) ecosystem services, among other extractable natural resources, and can thus contribute to responsible resource management [7]. Such considerations have underpinned the importance for incorporating geodiversity into geoconservation decisions onshore [8,9]. Recent studies explicitly include marine geoheritage in underwater research, which is an underrepresented topic as well [3]. Synergy between land-based and underwater geoheritage research methods can strengthen the relations that exist between terrestrial and marine sites [10]. For these reasons, we believe that mapping geodiversity below sea level can support management in marine environments. The term geodiversity is used in different contexts for submarine landscapes (or 'seascapes'). Kaskela et al. [11], for example,

refer to "the variation and/or patchiness of benthic substrates and seabed features" in their research on the role of geodiversity in Finnish shelf ecosystems. Zelewska et al. [12] mention "the diversity of abiotic components of the natural environment of a seascape" in relation to a formerly glaciated boulder area on the southern Baltic Sea shelf. In the deep ocean, the mineralization near local hydrothermal vents along spreading ridges is addressed in a study by Fouquet et al. [13] as a complex 'diversity' process. Mattos et al. [14] use geodiversity in relation to emerged portions of oceanic islands and to coastal environments. On a global scale context [15], Cherkasov considers mid-oceanic ridges and island arc systems as a first level of geodiversity in oceanic settings, whereas, e.g., the variety of hydrothermal vents is considered as a second level of geodiversity and thus part of a higher hierarchical level. Submarine geomorphological features such as seamounts, canyons, basins and slopes are known to provide habitat types that host unique biodiversity on the ocean floor [16]. On the geomorphological map of the sea floor of all oceans [17], such abiotic habitat types are regarded as features identified as part of ocean floor geodiversity. Although a definition of geodiversity for ocean floor settings seems to be adjusted for coastal and submarine settings, we here mention the commonly used definition of geodiversity by Gray [1]. This definition was developed for the terrestrial environment: "the natural range (diversity) of geological (rocks, minerals, fossils), geomorphological (landform, processes), soil and hydrological features. It includes their assemblages, structures, systems and contributions to landscapes". Following this definition, only few global-scale terrestrial geodiversity studies have been published, using harmonized and openly available environmental data layers. Muellner-Riehl et al. [18] used a global geodiversity index, based on harmonized, accessible input layers (geology, soils, hydrology, and topography) to explain biodiversity variation in mountains across the world. Polman et al. [19] used global geodiversity layers to explain the differences in geodiversity of approximately 150 UNESCO Global Geoparks in comparison to randomly selected samples across the remaining parts of the terrestrial part of the globe.

To broaden the land-based geodiversity research to offshore areas and to accommodate the transferability of our geodiversity assessment method to ocean floor regions, we use the following layers in our geodiversity index mapping: sediment thickness distribution (as a subaqueous equivalent for terrestrial soil variation), ocean floor geomorphology and bathymetry data. Whereas terrestrial geodiversity resources, such as oil, gas, (brown) coal, minerals and rare earths have been extracted extensively since the start of the industrial revolution, mineral resources of the ocean floor have only recently gained the attention of many countries and private companies [20] as alternatives for traditional mining to facilitate and ensure the transition to a circular and green economy [21]. The demand for resources such as metals, rare earths and other compounds is rising steeply. Known terrestrial occurrences of minerals and metals are rapidly being depleted [22], and are sometimes exclusively available in conflict areas (e.g., cobalt) where implementing corporate social responsibility is difficult [23]. Estimates show that the ocean floor contains more reserves of nickel (Ni), cobalt (Co), and manganese (Mn) than the current resources on land, while substantial copper (Cu), lithium (Li) and silver (Ag) resources have economic potential [22,24,25]. This potentially large amount of minerals and metals has not been extensively commercially explored and extracted yet [25], mainly because of the inherent challenge of exploration and mining at large water depths.

Currently, three key mineral aggregates widely occur on the deep-sea floor that provide potential alternatives to onshore resources. They are usually distributed in areas below 200 m water depth [20]: (1) polymetallic nodules; (2) sea floor massive sulphide; and (3) cobalt-rich crusts. Polymetallic nodules (PN) are deposits mostly composed of up to 12 cm large manganese and iron lumps [26], with distinct traces of Ni, Cu, Co, molybdenum (Mo) and rare earth elements (REEs) [25,27,28]. Occurrences of PN are reported at water depths between 3000 and 6000 m [27,29] in water-rich sediments and in basaltic substratum overlain by sediment [28,30]. PN take several million years to form [25,27], typically under low sedimentation rates in abyssal environments [28].

Sea floor massive sulphide (SMS) deposits were first discovered in association with hydrothermal vents in the late 1970s [15]. They are most abundant at depths of 800–4000 m around hydrothermal vents [27,29]. Hydrothermal vents occur along mid-oceanic ridge systems across all oceans and in island arc settings and are relatively young on a geological time scale [24,28]. The spreading rate at mid-ocean ridges influences the amount of SMS formed. At relatively low spreading rates < 4 cm/year, 80% of SMS is formed, while formation diminishes to only 2% above spreading rates of 8 cm/year [31]. This high degree of fine-scale variation is typical for spreading environments, and is characterized by magnetic anomalies and temperature variations around active hot vents and the development of black smoker chimneys [14,32]. Spreading environments can therefore be regarded as highly geodiverse environments.

Cobalt-rich crusts (CRCs) mainly contain Mn, Fe, Co, Cu, Ni and platinum (Pt) [24,28,29]. CRCs are known to occur on seamounts [26,28] and on ridges and plateaus [24,28]. They generally occur at water depths between 400 and 7000 m [29,30]. The increase in crust thickness (up to 25 cm) is a very slow process and is thought to be directly linked to seafloor age [28].

Currently, the International Seabed Authority (ISA) has the responsibility to ensure the protection of the marine environment from harmful effects that may arise from deep-seabed-related activities [32]. The mineral resources of the deep seabed are designated as the common heritage of all mankind under the United Nations Conventions on the Law of the Sea [32]. In 2017, the UN proclaimed the Decade of Ocean Science for Sustainable Development (2021–2030) with the objective to promote sustainable ocean management and highlight the need for ocean observation and ocean research [33]. Although licenses have been granted to countries and private companies by the ISA for extracting the rich mineral resources in the Pacific Ocean, little is known about the impact of seabed mining on ocean floor geodiversity. Despite few impact assessments, schemes have been developed and applied [34,35]; the mention of geodiversity, however, is uncommon. Mining can directly cause the mortality of fauna, destroy habitats, lead to habitat fragmentation, and can potentially release toxic substances and alter hydrological conditions [26,36–38]. It is difficult to assess the scale and severity of such potential impacts, as possible changes are mostly hidden from direct observations. Moreover, the amount and capabilities of remote sensing sensors available for deep-sea research are relatively limited compared to those available for terrestrial monitoring.

In contrast, there is a rising number of studies on the relationship between geodiversity, biodiversity and habitat variability in the terrestrial realm [5] that use descriptive and (semi-)automated analyses techniques [39,40]. It is likely that seafloor geodiversity and its relation to ocean floor biodiversity hold similar relationships. Unique ecosystems with a rich biodiversity are present, for instance, on hydrothermal vents, seamounts, abyssal plains and back arcs [28]. Many submarine features, such as canyons, act as conduits for particles from the fertile coastal zones, and shelves to the deep basins fuel deep faunal communities [41]. These spatial relationships underpin the importance of their geosystem functions and suggest that the disturbance of geodiversity can harm biodiversity as well.

Given that little is known about the potential impacts of mineral extraction, we consider it important to map the geodiversity of the ocean floor using open access, harmonized data. By combining different geodiversity components, it is possible to generate an ocean floor geodiversity index to facilitate quantitative analysis. The resulting geodiversity index map can then be used to identify what components of geodiversity are affected and where, in relation to the extraction of mineral resources. With these applications in mind, we aimed to (1) develop an ocean floor geodiversity index map of the SW Pacific Ocean based on openly accessible input layers, and (2) analyse how the distribution of PN, SMS deposits and CRC was related to the geodiversity components, their indexes, and to the age of the ocean floor.

The study area was located in the SW Pacific Ocean between 20° N 143° E and 140° W 30° S, eastward of Papua New Guinea and Australia, covering the Coral Sea, Salomon Sea,

Bismarck Sea and large parts of the Indian Ocean. Numerous islands and small island states are located within this extent, including New Caledonia, the Solomon Islands, Fiji, Tuvalu, Vanuatu, Kiribati, Micronesia, the Marshall Islands and Samoa, among others. We selected this part of the Pacific Ocean because of its diverse oceanographical, geological, and geomorphologic setting, large water depth range, and known potential mineral resources. For these reasons "this region contains an incredible amount of geodiversity" [24]. The area covers parts of three major tectonic plates, the continental Indo-Australian and the marine Pacific and New Hebrides Plates [42]. Along the two plate margins, young ocean basins have formed comprising several micro-plates [24]. These processes have created a high degree of ocean floor geodiversity, characterized by ocean trenches, island arcs, ocean plateaus, basins, seamounts, seamount chains, ocean plumes, and rifted, submerged, aseismic, continental fragments [24].

## 2. Materials and Methods

We used five openly accessible data layers (Table 1) for the development of an ocean floor geodiversity index. In the literature, these datasets are indirectly described in relation to geodiversity components and partly to the spatial distribution of the three mineral resources. The datasets are comparable to data which are used in terrestrial index-based geodiversity assessments [18,43], although differences exist. Here, we combined a geomorphological map of the ocean floor and a bathymetry layer, so that a variety of geomorphological units, ocean floor roughness and water-depth variation would be reflected in the geodiversity index. We used a sediment thickness layer as a marine alternative for soil types. The layer acted as a record of the accumulation of marine sediments over time and was derived from data on, e.g., turbidity currents, transport by ocean currents and local submarine sources. A data layer of the ocean floor age was used to evaluate the influence of time on the formation of the mineral resources. The age information, however, was not included in the calculation of the geodiversity index.

**Table 1.** List of datasets, short descriptions and sources.

| Dataset | Description | Source |
|---|---|---|
| Seafloor geomorphology | Seafloor geomorphic polygon features (scale 1:500.000) | [17] http://www.bluehabitats.org/?page_id=58 (accessed on 2 June 2022). |
| Bathymetry | Ocean floor bathymetry raster (15 arc s) | [44] https://www.gebco.net/data_and_products/gridded_bathymetry_data/#global (accessed on 2 June 2022). |
| Sediments | Map of the distribution of sediment thickness | [45] https://www.ngdc.noaa.gov/mgg/sedthick/index.html (accessed on 2 June 2022). |
| Mineral deposits | Map of the distribution of PN, SMS and CRC | [32] https://www.isa.org.jm/contractors/exploration-areas (accessed on 2 June 2022). |
| Age | Raster (.nc format) of ocean floor age (2 arcmin) | [46] https://ngdc.noaa.gov/mgg/ocean_age/ocean_age_2008.html (accessed on 2 June 2022). |

### 2.1. Input Data

For the development of the geodiversity index and the analysis, we used the digitally available 1:500.000 scale vector-based geomorphological map of the ocean floor compiled by Harris et al. [17]. It depicts four levels of the main geomorphic base features. These four levels were the Shelf, Slope, Abyssal and Hadal zones, which were overlain by a 'classification' layer and a 'feature' layer (Table 1). The classification layer consisted of a Shelf classification (high, medium, low) and Abyssal classification (mountains, hills, plains). The feature layer included Escarpments, Basins, Reefs, Canyons, Guyots, Seamounts, Bridges, Sills, Glacial Troughs, Shelf Valleys, Rift Valleys, Troughs, Ridges, Spreading Ridges, Fans, Rises, Terraces, Trenches and Plateaus. We selected all the components of the

feature layer (except for Reefs, because we focused on water depths larger than 200 m) for use in the geodiversity index.

A bathymetric gridded data layer for the extent of our research area was downloaded from the GEBCO download facility as a 2D netCDF grid file (Table 1). The GEBCO_2020 Grid is a continuous, global terrain model for ocean and land with a spatial resolution of 15 arc s [44]. The bathymetric layer was used to calculate an ocean floor roughness layer.

The marine sediment thickness data were contained in a global dataset (the 'GlobSed' dataset, Table 1), which was constructed by compiling gridded sediment thickness data [45]. The 'GlobSed' map was constructed at 5 arcmin spatial resolution and was merged and updated from data derived from the NE Atlantic and Arctic oceans, the Mediterranean and the Southern Ocean regions. Thicknesses varied from zero to >750 m without any differentiation in grain size composition.

A dataset of ocean floor age (Table 1) has been published by Müller et al. [46]. Their data includes age uncertainty, spreading rates and spreading asymmetries of the world's ocean basins as geographic and Mercator grids with 2 arcmin resolution (pixel-registered). The age reconstruction is determined by linear interpolation between adjacent seafloor isochrons in the direction of spreading.

The mineral deposits input (Table 1) is a map of point locations of PN, SMS and CRC for the Southwest Pacific Ocean, derived from the ISA. The locations correspond to the original locations of grab samples of unconsolidated sediments stored in the ISA data repository.

### 2.2. Workflow for Ocean Floor Geodiversity

Our workflow (Figure 1) followed a semi-quantitative geodiversity index procedure [39], in which expert-based input layers and a derivative layer from a bathymetric grid were combined. The method builds on earlier methods described by, e.g., Pereira et al. [47] for Brazil and by Seijmonsbergen et al. [43] for the Hawaiian archipelago, and is separated into three routines: (1) data collection, (2) analyses and (3) deliverables.

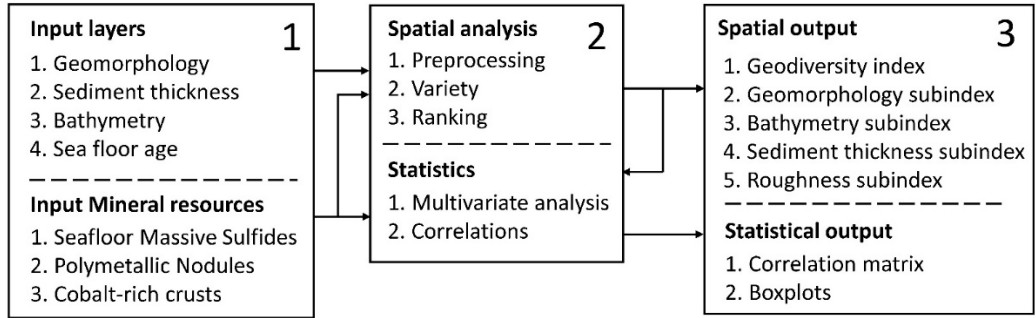

**Figure 1.** Workflow consisting of three routines: data collection (geodiversity components and mineral resources), analysis (spatial and statistical tools) and deliverables (maps and tabular data).

In the first routine, the input layers were downloaded from openly available data repositories (Table 1) and pre-processed to facilitate raster-based analyses in ArcGIS Pro 2.8 in the second routine. We scanned and digitized the sediment thickness and mineral resources maps and re-projected, rasterized and clipped all input layers to the analysis extent. An analysis grid of 2500 × 2500 m was created, guided by the work of Hengl [48], who provides a tool to calculate the finest, coarsest and recommended grid cell size based on a given a set of input layers with known resolution and/or scale. We decided to apply the coarsest resolution of 2500 × 2500 m for our analysis grid and a raster cell size of 250 × 250 m for all input layers, which best accommodated the zonal statistical analyses and which, at the same time, was computationally efficient. Sub-indices were calculated for each component and reclassified into 5 categories (very low, low, medium, high, very high), based on 'Jenks natural breaks classification' method, available in ArcGIS Pro 2.8. The bathymetry model was used to calculate a bathymetry sub-index, for which we classified

the water depth per 250 × 250 m grid cell into 5 categories, ranging from very shallow (1) to very deep (5), using the 'Jenks natural breaks classification' method. A similar approach was used for the ocean floor roughness, which was based on the water-depth range per analysis grid cell (very low was class 1, very high was class 5). The variety per analysis grid cell for the geomorphology was assigned ranks from very low (class 1) to very high (class 5). The sediment thickness layer was subdivided into five classes from thin sediment thickness (very low, class 1) to thick (very high, class 5). The geodiversity index then adds the scores of the four sub-indices (geomorphology, bathymetry, roughness and sediment distribution) and calculates total geodiversity scores per grid cell. The total scores are then subdivided into five categories, ranging from very low (class 1), low (class 2), medium (class 3), high (class 4) to very high (class 5). The final geodiversity index for the ocean floor is expressed as:

$$GDi = Gi + Sri + Bi + Ssi \tag{1}$$

where GDi = geodiversity index; Gi = geomorphology index; Sri = seafloor roughness index; Bi = bathymetry index; and Ssi = seafloor sediment thickness index.

### 2.3. Statistics Analysis

We calculated basic statistics (min, max and median) to find relations between the geodiversity components (bathymetry, sediment thickness, seafloor roughness) and ocean floor age and the three mineral resources. The number of samples ranged from 46 for SMS to 308 for CRC and 327 for PN. The 'extract multi values to point' tool was used for each location of a mineral resource to extract geodiversity information and the 'band collection statistics' tool was used for the multivariate analysis. Matlab [49] was used to calculate correlations between mineral resources, the GDi and its components, and ocean floor age. We decided not to include the seafloor age in the GDi because a substantial part of the seafloor off the coast of east Australia belongs to the continental Indo-Australian Plate, for which ages are largely missing. Moreover, microplates that were formed in the collision zone of the Indo-Australian and Pacific Plates [24] caused local anomalies in the age relationship with increasing distance from the spreading ridges.

### 3. Results

#### 3.1. Index Maps

The five resulting sub-index maps are displayed in Figure 2 (maps A–E) and the final geodiversity index map of the Southwest Pacific Ocean floor is shown in Figure 2 (map F). The geomorphological diversity (Figure 2A) was dominated by very low and low diversity classes and reflected broad plateau areas and deep-ocean basins. However, a high diversity of submarine landforms did occur, mostly in active margins, trenches and subduction zones. They showed as small clusters or isolated cells within the larger homogeneous submarine landforms, or were not identified by the index because they were too small in comparison to the grid cell size. Large parts of the ocean floor classified as very low and low roughness (Figure 2B). Roughness variation was prominent in active plate boundaries, island arcs and submarine volcanic island hotspots where slope steepness and elevation variation can change over relatively short distances. This also implies that roughness variation of the ocean floor within 2500 × 2500 m cells is restricted, especially in geologically stable environments, such as deep hadal zones or elevated submarine plateau areas, or areas that have been flattened by secondary processes, such as the accumulation of sediments. The sediment thickness map shows that the continental plate contained the thickest accumulation of sediment (Figure 2C), whereas deeper and more remote submarine regions were covered by relatively thin unconsolidated deposits.

The bathymetry index (Figure 2D) had very low/low values on the continental plate off the coast of Australia because of its relatively shallow water depths. Active subduction zones, in contrast, were characterized by higher index scores, whereas actively uprising islands and plateau areas were characterized by lower bathymetry index scores.

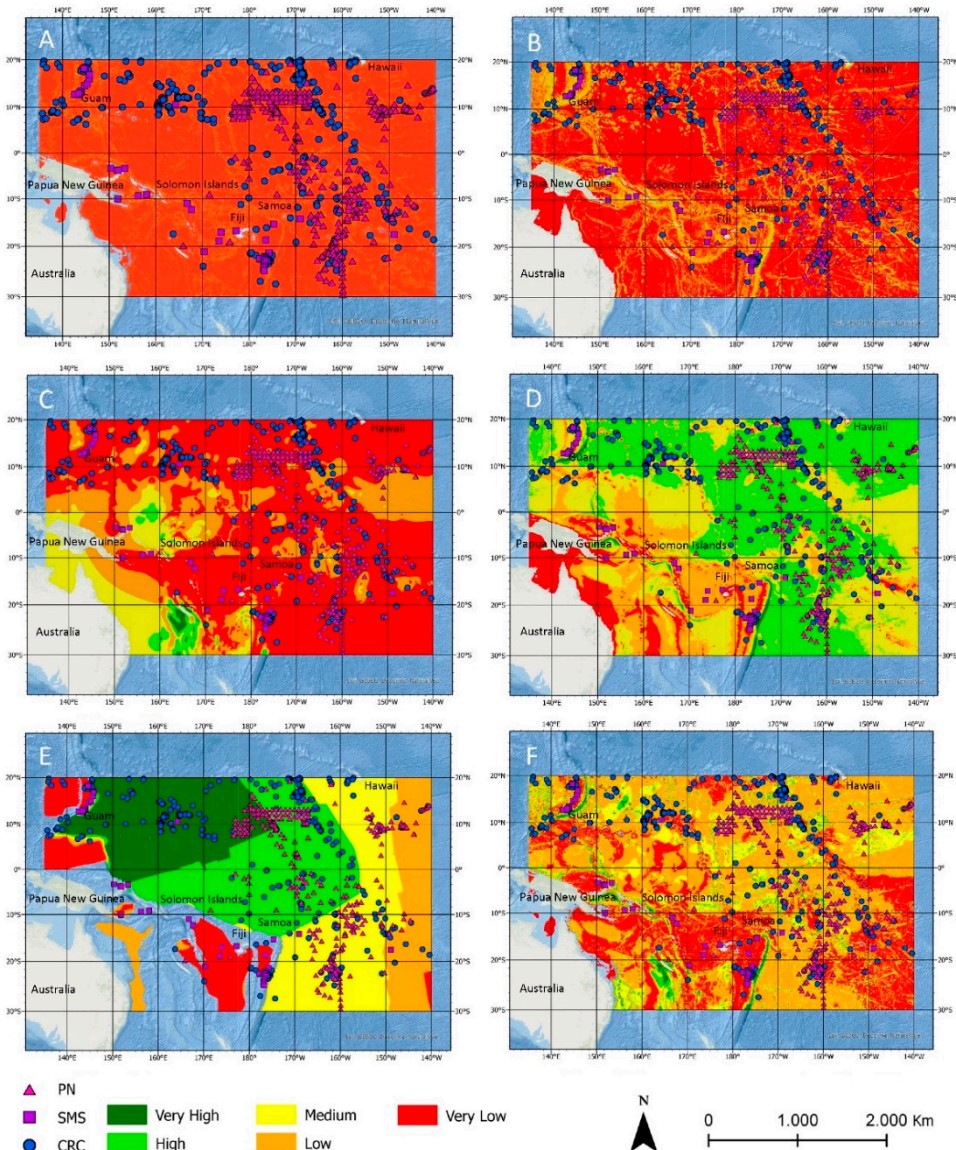

**Figure 2.** Sub index maps of (**A**) geomorphology, (**B**) seafloor roughness, (**C**) sediment thickness, (**D**) bathymetry, (**E**) ocean floor age and (**F**) geodiversity index map.

The oldest ocean floor ages (Figure 2E) occurred near the continents, generally away from the central parts of the ocean and towards the continental margins, as a consequence of marine plate tectonic motion. This suggests that, over time, the contribution of sinking marine plates increases, which is in line with the seafloor age raster (which was used separately). No indications of young ocean floor ages were found around the many volcanic hotspots, which was probably due to the limited number of age measurements and/or the relatively low resolution of the original dataset. Another anomaly is that parts of the Australian continental plate were missing; however, this did not affect our analysis because of the rare occurrences of mineral resources in this region.

The GDi map is the result of the combined total scores of the indices presented in Figure 2 (maps A–D). 'Hotspots' (areas with high and very high index scores) of geodiversity are mainly located in deep-sea trenches and subduction zones, and reflect combinations of high roughness and relatively large depth variation over short distances.

*3.2. Statistical Analysis*

　　　Correlations between the sub-indices and the mineral occurrences are presented in Table 2. The distribution of PN showed strong negative correlations with the geodiversity index (GDi was −0.97) and all individual components (−0.88 to −0.35). Seafloor age was moderately positively correlated with the distribution of PN. In contrast, the distribution of SMS had a weak positive correlation with the geodiversity index (0.38) and a moderately strong correlation with the bathymetry index (0.66). Mostly, SMS occurrence was associated with relatively shallow depths at spreading ridges, i.e., surroundings which were relatively geodiverse.

**Table 2.** Correlations between the three mineral resources and the geodiversity index and its components. GDi = geodiversity Index; Sri = seafloor roughness index; Bi = bathymetry index; Ssi = seafloor sediment index; Gi = geomorphology index; and Sai = seafloor age index. Mineral resources: CRC = cobalt-rich crust; SMS = sea floor massive sulphide; and PN = polymetallic nodules.

| Sub-Indices | GDi | Sri | Bi | Ssi | Gi | Sai |
|---|---|---|---|---|---|---|
| GDi | 1 | | | | | |
| Sri | 0.52 | 1 | | | | |
| Bi | 0.75 | −0.18 | 1 | | | |
| Ssi | 0.53 | −0.15 | −0.44 | 1 | | |
| Gi | 0.25 | 0.22 | −0.03 | −0.05 | 1 | |
| Sai | −0.10 | −0.01 | 0.14 | 0.03 | 0.01 | 1 |
| **Mineral resources** | | | | | | |
| CRC | −0.03 | −0.98 | 0.32 | −0.78 | −0.78 | 0.88 |
| SMS | 0.38 | −0.80 | 0.66 | −0.78 | −0.76 | −0.71 |
| PN | −0.97 | −0.88 | −0.35 | −0.77 | −0.74 | 0.54 |

　　　The geodiversity index was positively correlated with the seafloor sediment thickness index (0.53) and the seafloor roughness index (0.52). CRC had a strong negative correlation with the seafloor roughness index (−0.98), the seafloor sediment thickness index (−0.78), and the geomorphological index (−0.78), while the correlation with the bathymetry index (0.32) was weakly positive. In general, CRCs occur in relatively flat to sloping areas characterized by low geomorphological variation (e.g., large plateaus, on top of guyots) and deeper water environments.

　　　The boxplots of Figure 3 show the relation between the range and median values of the three mineral resources, on the one hand, and three components of geodiversity and the ocean floor age on the other. The distribution of PN, SMS and CRC across bathymetric depth showed distinct differences. CRC samples (*n* = 308) typically had a wide depth range, but were concentrated at intermediate depths and occurred predominantly between −5732 m and −2575 m (median = −4065 m), with only eight data points at depths shallower than −1000 m. The SMS samples (*n* = 46) were distributed between −406 m and −4077 m depth with a median of −2382 m. The PN samples (*n* = 327) were distributed between −6344 m and −1535 m depth, with a median depth of −5183 m. Only seventeen data points occurred above −4000 m, which indicates that the majority of the PN samples were distributed in the deeper parts of the ocean floor. In general, the bathymetrical component of geodiversity seems to be a good predictor to separate the spatial distribution of PN and CRC from SMS.

　　　The CRC samples were from relatively old parts of the seafloor tectonic plates. The age range was between 15 My and 175 My with a median age of 120 My. For 28 out of 308 data points, the ocean floor age data were not available, mainly because they were located on the continental Indo-Australian plate. SMS samples typically occurred on relatively young parts of the ocean floor, with ages less than 32 My and a median age of 5. The median age

of the ocean floor of PN was 112 My, with a youngest age of ~14 My (only one point) and with the majority of the data points between 53 My and 158My. The seafloor age was found to be a good predictor of the separation of SMS from CRC and PN.

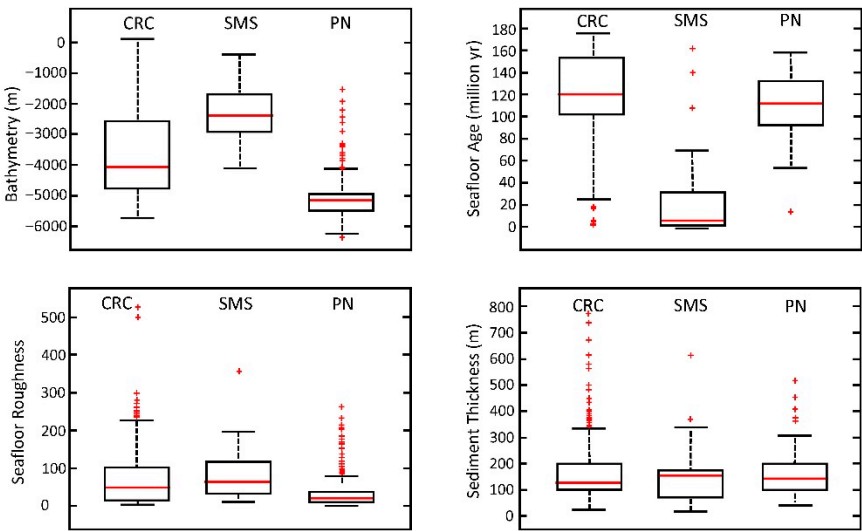

**Figure 3.** Boxplots presenting the range and median values of occurrences of CRC = cobalt-rich crusts, SMS = sea floor massive sulphide, and PN = polymetallic nodules, in relation to bathymetry (top left), seafloor age (top right), seafloor roughness (lower left) and sediment thickness (lower right).

The seafloor roughness revealed less distinct variations, although SMS occurred in ocean floor environments with slightly higher roughness, while CRC and PN occurred in environments with relatively lower roughness. In general, all three resources were concentrated at relatively low values of seafloor roughness (<100). The maximum roughness values were 526 352 and 257 for CRC, SMS and PN, respectively. Seafloor roughness did not distinctly separate SMS from CRC and PN.

In general, most samples were derived from sediment thicknesses between 100 and 200 m, and CRC samples were also taken from locations with >750 m sediment thickness. The CRC sample locations had thicknesses between 19 m and 774 m, with a median value of 128 m, while SMS sample locations had sediment thicknesses ranging between 13 m and 611 m, with a median of 152 m. PN samples were mostly taken from sediments with thickness ranging between 40 m and 517 m with a median of 141 m.

## 4. Discussion

We developed a geodiversity index map for the Southwest Pacific Ocean based on openly available data. We calculated correlations of the geodiversity index, its components, seafloor age and the distribution of SMS, PN and CRC. The resulting maps and relationships provide new insight into the extent and location of geodiversity variation across the Southwest Pacific Ocean floor and of the geodiversity of potential mining resources.

### 4.1. Geodiversity Definition and Index Development

For the development of the geodiversity index, we used openly available, relatively low-detail datasets. This secures transparency, reproducibility and transferability in future index-based geodiversity analyses of other ocean floor environments, which is necessary for making objective comparisons between different areas or with biodiversity [7]. Geodiversity is often used in relation to protected areas [50] in terrestrial settings such as UNESCO Global Geoparks [51], Natura 2000 areas in Europe [52] and World Heritage Sites, but less frequently to marine protected areas [53] or other marine conservation areas. A clear definition will contribute to embed geodiversity in applied submarine research, activities and conservation practice. We therefore adhere to the flexible globally valid definition used

by UNESCO [54], which is found on their website, and promote the recently established 'International Geodiversity Day', which was introduced on 6 October 2021: "Geodiversity is the natural portion of the planet that is not alive both at the surface and in the interior. It includes the Earth's minerals, rocks, fossils, soils, sediments, landforms and topography, as well as hydrological features such as rivers and lakes and the processes that make and modify these features". This definition is broader than that of Gray [1] and allows the flexible use of input data for semi-quantitative studies at multiple scales. For example, high-resolution bathymetric datasets can be used in detailed case studies as surrogates in the absence of fine-scale geomorphology to explain benthic habitat distribution [55]. Or, a combination of a detailed expert-based geomorphological map, a bathymetric relief energy map and a sediment texture class map could be used to evaluate underwater geodiversity [12]. It is well known that volcanic islands evolve over time and can eventually become part of the ocean floor [56]. As a consequence, terrestrial geodiversity will transform into marine geodiversity (e.g., a volcanic island develops into an extinct guyot). Therefore, efforts should be made to further explore the temporal dynamics of geodiversity on the interface of land and sea. Based on the results of our study, we believe that the increased resolution of future layers (datasets) in geodiversity-related research of the ocean floor will generate further insight into the relations between bathymetry, ocean floor geomorphology and sediment distribution. For example, the identification of seamounts based on automatic morphometric analysis techniques [57] applied to high-resolution bathymetric data can add details to geodiversity patterns of the ocean floor. Moreover, the relations to mineral resource distribution should be further improved or reconsidered in future studies. This belief is based on the vast increase in commercial organizations and nations that, over the last 5–10 years, have applied for new contracts to explore and extract mineral resources [28,32] and which seek to practise responsible the exploration of mineral resources to minimize environmental impact. Deep-sea mining policies aim to develop and protect the seabed for mankind; therefore, it is suggested that regulations for mining are defined, such as the exchange of information, impact assessments for seabed ecosystems, assurances of technological capacity to monitor key environmental parameters and measure change, and effective operating procedures for accident responses [58].

### 4.2. Geodiversity and Distribution of Mineral Resources

The distribution of the three mineral resources can be explained, for a major part, by the geodiversity of the ocean floor environment and the context of their formation history. For example, sea floor age has a moderately positive correlation with PN distribution. This is understandable, given that nodules take millions of years to form [25,27] at the conditions present on the deep ocean floor, i.e., stable plateaus and deep hadal areas, under low sedimentation regimes and at slow plate movement rates, away from active tectonic zones. In contrast, SMS develops in relatively young, shallow, tectonically active, geodiverse environments (e.g., spreading ridges), in which sedimentation is still restricted and roughness is found at a fine scale; thus, the available geomorphological data are relatively underrepresented in the assessment. This type of knowledge is not always propagated in the geodiversity index and its components with the current quality of the input data, in particular with regard to the scale, cell size used, and the details identified in the compiled thematic map.

All samples were collected in areas covered by unconsolidated deposits, but this does not imply that the mineral resources were absent in rocky substrates. In our analysis, the sediment thickness relations were comparable for SMS, CRC and PN, but the material composition (clay, silt, sand) was not taken into account. Other influencing factors, such as distance from the source, ocean currents and chemical composition of the sediments may turn out to be useful parameters in follow-up research.

*4.3. Concluding Remarks*

This is an initial explorative study of the geodiversity of the Southwest Pacific Ocean floor which relates geodiversity and its components to the distribution of mineral resources. The results show that the ocean floor environment and the time necessary for the formation of the three mineral resources are relate to the bathymetry component of geodiversity and to the age of the ocean floor, and to a lesser extent to seafloor roughness, geomorphology and sediment thickness. From a geodiversity perspective, the ocean floor is still an underrepresented environment, mainly because the deep remote parts of the ocean floor are difficult to access (both physically and virtually). Interdisciplinary research is imperative to harmonize the interests of marine biodiversity, ocean floor geodiversity and economic activity. For example, the rich archives of drilling programs can be included in geodiversity assessments, along with other marine data derived from sonar, side scan or seismic survey methods. Further research and development of the method should aim at optimizing the method by generating and applying fine-scaled datasets, reducing uncertainty and error propagation, and testing the method on other parts of the ocean floor and shallow marine shelf seas.

**Author Contributions:** Conceptualization, A.C.S.; methodology, A.C.S., S.V., L.W.; validation, S.V. and L.W.; formal analysis, S.V. and L.W.; investigation, S.V., L.W. and A.C.S.; data curation, S.V. and L.W.; writing—original draft preparation, A.C.S.; writing—review and editing, A.C.S., S.V., L.W. and K.F.R.; visualization, S.V. and L.W.; supervision, A.C.S. and K.F.R. All authors have read and agreed to the published version of the manuscript.

**Funding:** This research was internally funded by the Theoretical Computational Ecology Group of the Institute for Biodiversity and Ecosystem Dynamics of the Universiteit van Amsterdam.

**Acknowledgments:** The GIS-studio (www.GIS-studio.nl (accessed on 2 June 2022)) of the Universiteit van Amsterdam is thanked for the use of soft- and hardware.

**Conflicts of Interest:** The authors declare no conflict of interest.

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
