# Peer review of "Exploring Ocean Floor Geodiversity in Relation to Mineral Resources in the Southwest Pacific Ocean"

_resources, doi:10.3390/resources11070060_

Round 1

Reviewer 1 Report

Dear Authors,

I've read your manuscript with a great pleasure. For already some time, I have expected the appearance of such a work devoted to deep-marine geodiversity, and now I'm very happy to see it is as good as the international research community needs really. The topic is very urgent, and the selected example is very representative. The manuscript is well-written and well-illustrated. In my opinion, it is very important that it is written simply and compactly – these features will facilitate its circulation among the experts. I have some recommendations, which I hope can make your work even stronger.

  • Introduction and subsection 4.1: I think you need to add notions of marine geoheritage. Several related works have been published recently – please, check literature (if you have access to "Scopus", you can easily find the works with the words "marine" and "geoheritage" in titles or abstracts).
  • Materials and Methods: I think you need to create a new subsection "Oceanographical and Geological Setting" (some information can be moved to there from Introduction).
  • Discussion: please, write a bit about the issues of deep-sea mining policy, co-existence of ocean-floor and island geodiversity, and ocean-floor geodiversity accessibility ("physical" or virtual). Of course, do not forget about citing the related works.
  • Discussion: I encourage you to think about the relation of the concept of ecosystem/geosystem services to the general problem you address (this may lead to far-going propositions, and, thus, I do not insist on doing this, although such a discussion would benefit your work).
  • Concluding remarks: may be to write about the potential usefulness of the data from the DSDP-ODP program and other initiatives of ocean-floor studies?

Good luck with revisions! I very hope to see your contribution published soon.

Author Response

Response to Reviewer 1 

Dear Authors,

I've read your manuscript with a great pleasure. For already some time, I have expected the appearance of such a work devoted to deep-marine geodiversity, and now I'm very happy to see it is as good as the international research community needs really. The topic is very urgent, and the selected example is very representative. The manuscript is well-written and well-illustrated. In my opinion, it is very important that it is written simply and compactly – these features will facilitate its circulation among the experts. I have some recommendations, which I hope can make your work even stronger.

Thank you for these nice words. We tried to include most of your raised issues, but aimed at keeping the text concise, not speculative, because we, indeed, see this as a first attempt to assess and address ocean floor geodiversity, from which other research may benefit and extend..

General answer: As suggested, we carefully repaired the style/language issues (raise by one of the other reviewers as well)

Below we point-by-point respond to your issues raised.

Introduction and subsection 4.1: I think you need to add notions of marine geoheritage. Several related works have been published recently – please, check literature (if you have access to "Scopus", you can easily find the works with the words "marine" and "geoheritage" in titles or abstracts).

Response

Thank you for this suggestion. Although our objective was initially not to include geoheritage in our paper we have included two more references in the introduction, because we agree that this is a recent development with respect to marine research and shared methods can improve our understanding of both environments. We inserted quotes and statement by Ruban (2021) and Coratza (2019) in the introduction to address geoheritage, which is underrepresented in marine studies as well: see new lines 43-47.

Materials and Methods: I think you need to create a new subsection "Oceanographical and Geological Setting" (some information can be moved to there from Introduction).

Response

We feel that "Oceanographical and Geological Setting" is not really part of our Materials and Method section. The materials we used, are well-known published datasets, the metadata info of these datasets is perfectly described in the referred publications, while content is addressed as well. We only give short summaries in our text. In addition, geology is not a component that we directly included in our analysis. Therefore we feel a separate section on that topic is not needed in the Method/Materials section, two other reviewers did not mention this issue at all. We do agree that this topic is relevant. We therefore added a few extra words in the introduction and now e.g. explicitly mention "Oceanographical and Geological Setting" in line 175/176.

Discussion: please, write a bit about the issues of deep-sea mining policy, co-existence of ocean-floor and island geodiversity, and ocean-floor geodiversity accessibility ("physical" or virtual). Of course, do not forget about citing the related works.

Response

We have extended our discussion with some lines devoted to island-based and ocean-floor geodiversity (lines 437-441, referring to Whittaker et al. 2017), mining policy (lines 452-457, referring to policies suggestions mentioned by Kirkham et al. 2020) and ocean floor geodiversity (lines 488-489, no reference needed here)

Discussion: I encourage you to think about the relation of the concept of ecosystem/geosystem services to the general problem you address (this may lead to far-going propositions, and, thus, I do not insist on doing this, although such a discussion would benefit your work).

Response

We understand your encouragement, however, we would rather not go into much speculation or not very well-supported statements or ideas in this manuscript. Therefore, we are happy to leave this open to future follow-up research (which we will continue on as well).

Concluding remarks: may be to write about the potential usefulness of the data from the DSDP-ODP program and other initiatives of ocean-floor studies?

Response

Good point! We cannot make any clear conclusions on the data from this program. However, we added some lines in the Concluding Remarks on the potential to use other data (lines 491-493)

Good luck with revisions! I very hope to see your contribution published soon.

Thank you!

Reviewer 2 Report

The paper, entitled "Exploring ocean floor geodiversity in relation to mineral resources in the southwest Pacific Ocean", relates geodiversity to the distribution of minerals  resources in sea floor. The paper is well written and well structured. The methodology adopted is adapted to the geological setting and is well described. The authors used an openly available relatively low-detailed datasets that facilitate the transferability in future index-based geodiversity analyses of other ocean floor environments. The overall quality of the aper is high and I propose to accept it to publication in the special volume.

Author Response

Thank you for your encouraging words!

Reviewer 3 Report

Dear authors,

It is an innovative work that brings a necessary and timely discussion. The choice of area is well justified, as is the method used to obtain the Geodiversity Index Map.

The theoretical basis presented is excellent and allows for discussing marine geodiversity and its components. It forces us to reflect on the ecosystem service of provision (Gray, 2013), including the relationship with marine biodiversity and the human need for resources. Excellent job. I have a few suggestions to make as follows:

Line 41 - Question: Do you consider it appropriate to use the term "sustainable" to manage the use of mineral resources from the point of view of exploitation? "Sustainability" is a term that presupposes the availability of resources for future generations. Wouldn't it be better to use "responsible" management?

Line 92 - The same discussion made in the previous item is valid for these arguments.

Figure 1 – item 3 – subindex

Line 303 – ocean

Line 347 - I suggest moving the paragraph to a position before Figure 3

Line 434 - components

Best regards.

Author Response

Response to Reviewer 3 Comments

Dear authors,

It is an innovative work that brings a necessary and timely discussion. The choice of area is well justified, as is the method used to obtain the Geodiversity Index Map.

The theoretical basis presented is excellent and allows for discussing marine geodiversity and its components. It forces us to reflect on the ecosystem service of provision (Gray, 2013), including the relationship with marine biodiversity and the human need for resources. Excellent job. I have a few suggestions to make as follows:

Line 41 - Question: Do you consider it appropriate to use the term "sustainable" to manage the use of mineral resources from the point of view of exploitation? "Sustainability" is a term that presupposes the availability of resources for future generations. Wouldn't it be better to use "responsible" management?

Response

Thank you for you for this remark. We actually used the term ‘responsible’ already in the discussion and fully agree to use responsible. We changed it here as well.

Line 92 - The same discussion made in the previous item is valid for these arguments.

Response

We agree (although ‘your remark to 'previous item’ is a bit general?) - we did not change anything here.

Figure 1 – item 3 – subindex

Response

We inserted an updated figure to consistently to write subindex and a few other minor typos 

Line 303 – ocean

Response

Adjusted

Line 347 - I suggest moving the paragraph to a position before Figure 3

Response

This might be a good idea – but the final editing and responsibility is done by the journal

Line 434 – components

Response

We adjusted this typo